# Pynapple, a toolbox for data analysis in neuroscience

Guillaume Viejo[1,2], Daniel Levenstein[1,3], Sofia Skromne Carrasco[1], Dhruv Mehrotra[1], Sara Mahallati[1], Gilberto R Vite[1], Henry Denny[1], Lucas Sjulson[4], Francesco P Battaglia[5], Adrien Peyrache[1]*

[1]Montreal Neurological Institute and Hospital, McGill University, Montreal, Canada; [2]Flatiron Institute, Center for Computational Neuroscience, New York, United States; [3]MILA – Quebec IA Institute, Montreal, Canada; [4]Departments of Psychiatry and Neuroscience, Albert Einstein College of Medicine, Bronx, United States; [5]Donders Institute for Brain, Cognition and Behaviour, Radboud University, Nijmegen, Netherlands

**Abstract** Datasets collected in neuroscientific studies are of ever-growing complexity, often combining high-dimensional time series data from multiple data acquisition modalities. Handling and manipulating these various data streams in an adequate programming environment is crucial to ensure reliable analysis, and to facilitate sharing of reproducible analysis pipelines. Here, we present Pynapple, the PYthon Neural Analysis Package, a lightweight python package designed to process a broad range of time-resolved data in systems neuroscience. The core feature of this package is a small number of versatile objects that support the manipulation of any data streams and task parameters. The package includes a set of methods to read common data formats and allows users to easily write their own. The resulting code is easy to read and write, avoids low-level data processing and other error-prone steps, and is open source. Libraries for higher-level analyses are developed within the Pynapple framework but are contained within a collaborative repository of specialized and continuously updated analysis routines. This provides flexibility while ensuring long-term stability of the core package. In conclusion, Pynapple provides a common framework for data analysis in neuroscience.

## eLife assessment

This paper introduces the python software package Pynapple and a separate package of more advanced routines (Pynacollada) to the Neuroscience/Neural Engineering community. Pynapple provides a set of data objects and methods that have the potential to simplify data analysis for neural and behavioral data types. This represents a **valuable** contribution to the field. With more examples and as a live coding notebook, the evidence was judged to be **compelling**.

## Introduction

The increasing size of datasets across scientific disciplines has led to the development of specific tools to store (*Folk et al., 2011*; *Wells and Greisen, 1979*), analyze (*Pedregosa, 2011*), and visualize (*Maaten and Hinton, 2008*) them. While various programming environments such as Matlab and R have long been commonly used in data science, Python has progressively become one of the most popular programming languages (*McKinney, 2011*). This is due to its open nature, large community-driven development, and versatility of usage. As with virtually all other scientific fields, neuroscience faced the challenges of handling and analyzing large datasets by rapidly developing a wide range of

**\*For correspondence:**
adrien.peyrache@mcgill.ca

specialized tools to deal with each of these types of data (*Abraham et al., 2014*; *Tadel et al., 2011*; *Oostenveld et al., 2011*; *Bokil et al., 2010*; *Garcia et al., 2014*; *Freeman et al., 2014*) and corresponding analyses.

In systems neuroscience, calcium imaging and high-density electrophysiology make it possible to simultaneously monitor the activity of an increasingly large number of neurons (*Stevenson and Kording, 2011*; *Urai et al., 2022*). Often, this is combined with simultaneous behavioral recordings. As in all other fields, this has required the development of specific pipelines to process (*Pachitariu et al., 2016*; *Pachitariu et al., 2017*; *Hazan et al., 2006*; *Fee et al., 1996*; *Harris et al., 2000*; *Yger et al., 2018*; *Mathis et al., 2018*; *Zhou et al., 2018*; *Mukamel et al., 2009*; *Romano et al., 2017*; *Kaifosh et al., 2014*; *Pnevmatikakis and Giovannucci, 2017*) and store (*Teeters et al., 2015*; *Rübel et al., 2022*) the data. Despite this rapid progress, data analysis often relies on custom-made, lab-specific code, which is susceptible to error and can be difficult to compare across research groups. While several toolboxes are available to perform neuronal data analysis (*Oostenveld et al., 2011*; *Bokil et al., 2010*; *Garcia et al., 2014*; *Freeman et al., 2014*; *Nasiotis et al., 2019*; *Zugaro, 2018*; *Ackermann et al., 2018*) (see *Unakafova and Gail, 2019*, for review), most of these programs focus on producing high-level analysis from specified types of data, and do not offer the versatility required for rapidly changing analytical methods and experimental methods. Users can decide to use low-level data manipulation packages such as Pandas, but in that case, the learning curve can be steep for users with low, if any, computational background.

The key challenge for scientific code is balancing the need for flexibility and stability. This is especially true of science because results should be reproducible (between labs, between the past and the future, and between different experimental setups) while keeping up with rapidly changing requirements (e.g., due to new kinds of data, theories, and analysis methods). To meet these needs, we designed Pynapple, a general toolbox for data analysis in systems neuroscience with a few principles in mind.

The first property of such a toolbox is that it should be object-oriented, organizing software around data. This makes the programming environment very efficient for data analysis, particularly in systems neuroscience where data streams can be of very different types. For example, to compute the rate of an event, one can write a function that takes an array of event times and divides the number of elements by the time between the first to the last event. However, this approach neglects to consider that the appropriate epoch in which to calculate the rate could start earlier, or end later, than the first or last event. Addressing these concerns requires another argument, which defines the boundaries of the epoch on which the rate should be computed. Overall, this approach is error prone. The epoch boundaries and event times must be stored in the same time unit and with the same reference (i.e., simultaneous time 0) and the rate function itself can be erroneously called with arrays storing another type of data. In contrast, an object which is specifically designed to represent a series of event times can ameliorate these concerns. For example, it can be created from a specific data loader that ensures proper definition of time units and support epochs (i.e., true beginning and end of the observation time). It will then be immune to the arithmetic operations that can change the values of a generic array (e.g., an addition that is misplaced in the code). Further, the object can be endowed with a rate property that is specifically written for this object, reducing the odds of a coding error. While this approach may discourage users who are not familiar with this type of coding, the benefit far exceeds the effort of learning object-oriented programming, especially if the naming of the methods and properties is explicit.

Another property of an efficient toolbox is that a small number of objects could virtually represent all possible data streams in neuroscience, instead of objects made for specific physiological processes (e.g., spike trains). This ensures that the same code can be used for various datasets and eliminates the need of adapting the structure of the package to handle rare or yet-to-be-developed data types. Then, these objects should then be able to interact via a small number of basic and foundational operations, which are sufficient for most analyses. This allows users to quickly write new code for new use-cases, and easily understand and adapt code written by others, as the same methods can be used for any kind of data.

The toolbox should be able to load common data storage types, and the flexibility to create loaders for future and custom/lab-specific data. It should also support the development of yet-unknown, lab-specific, and specialized analysis methods. In other words, the customization of the

package to adapt to any dataset should happen at the input stage and the development of high-level analytical methods should take place outside the core package. The properties listed above ensure the long-term stability of a toolbox, a crucial aspect for maintaining the code repository. Toolboxes built around these principles will be maximally flexible and will have the most general application.

In this paper we introduce the Python Neural Analysis Package (Pynapple), designed with these axioms in mind. The core of Pynapple is five versatile time series objects, whose methods make it possible to intuitively manipulate and analyze the data. We show how Pynapple can be used with most raw neuroscience data types to produce the most common analyses used in contemporary neuroscience. Additionally, we introduce Pynacollada, a collaborative repository for higher-level analyses built from the basic functionality provided by Pynapple. A complete neuroscience data analysis pipeline using a common language supports open and reproducible code. As all users are also invited to contribute to the Pynapple ecosystem, this framework also provides a foundation upon which novel analyses can be shared and collectively built by the neuroscience community.

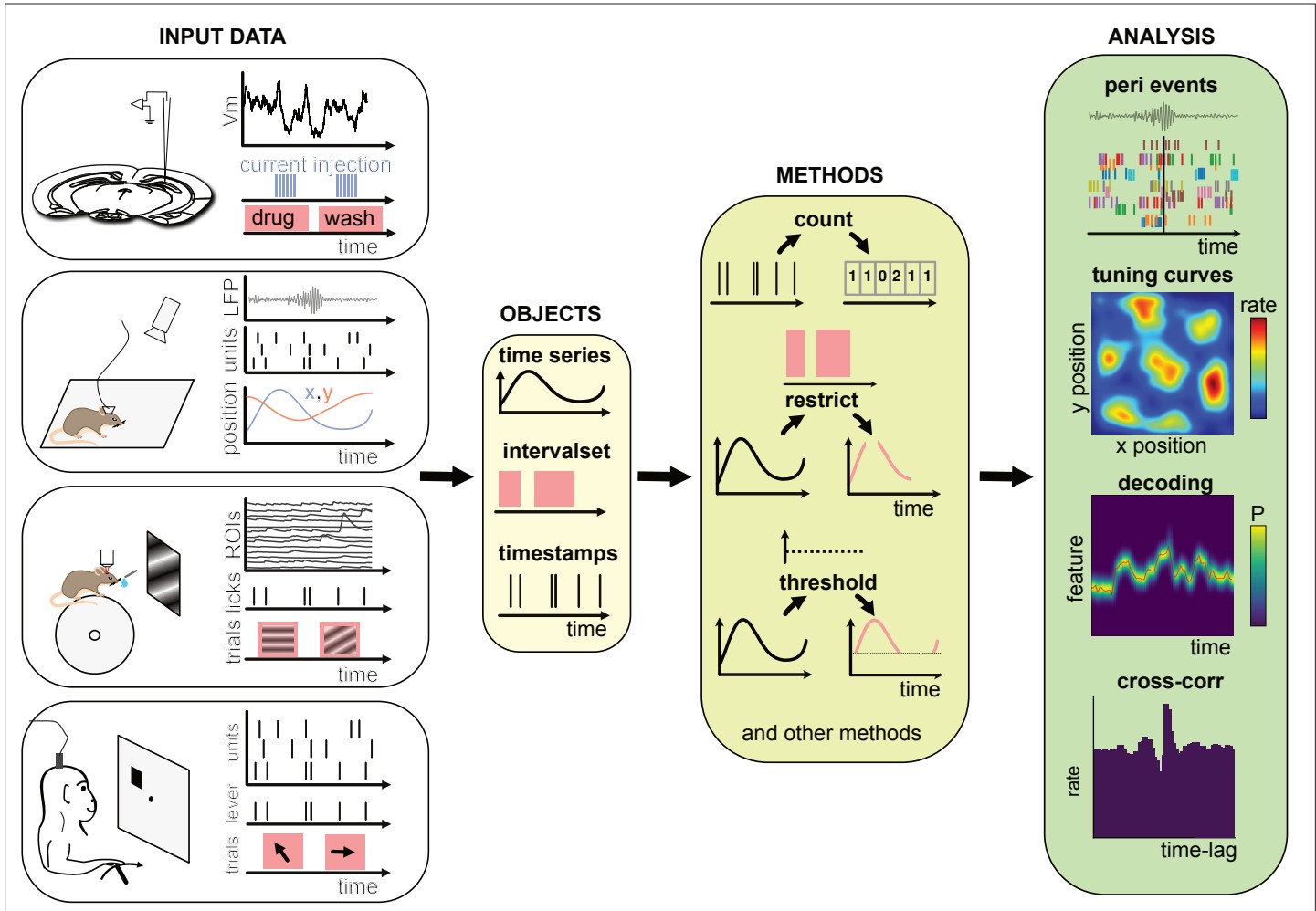

**Figure 1.** Data analysis with the Pynapple package. *Left*, any type of input data can be loaded in a small number of core objects. For example (from top to bottom): intracellular recordings in slice during which current is injected and drug is applied to the bath solution; extracellular recordings in freely moving mice whose position is video-tracked; calcium imaging in head-fixed mice during presentation of different visual stimuli and delivery of precisely timed rewards; extracellular recordings in non-human primates during the execution of cognitive tasks. *Middle*, object-specific methods allow the user to perform a wide variety of basic operations and to manipulate the data manipulations. *Right*, at a higher level, the package contains a set of foundational analysis methods such as (from top to bottom) peri-event alignment of the data (top), 1- and 2D tuning curves, 1- and 2D decoding; auto- and cross-correlation of event times (e.g., action potentials). These methods depend only on a few, commonly used, external packages.

## Results

### Core features of Pynapple

At its core, Pynapple is object-oriented. Because objects are designed to be self-contained and interact with each other through well-defined methods, users are less likely to make errors when using them. This is because objects can enforce their own internal consistency, reducing the chances of data inconsistencies or unexpected behavior. Overall, object-oriented programming is a powerful tool for managing complexity and reducing errors in scientific programming. Pynapple is built around only five objects that are divided into three categories: two objects represent event timestamps (one or several), two represent time-varying data (one or several time series at the same sampling times), and one represents time epochs. Raw or preprocessed data are loaded into these objects in the coding environment (*Figure 1*). The data loaders ensure that all loaded objects have the same time base. Hence, once objects are constructed, the user does not have to remember properties of the data such as the sampling frequency or alignment of data indices to clock time. Then, these objects can be manipulated with their own methods (i.e., object-specific functions). A large majority of data manipulations needed for most users can be achieved with a small number of methods. From there, Pynapple offer some foundational analyses, such as cross-correlation of event times. On top of this, the user may write analytical code that is project specific.

The most basic objects are timestamps (*Ts*), which are typically used for any discrete events, for example spike or lick times. The timestamped data (*Tsd*) object holds timestamps and associated data associated with each timestamp. For example, this object is used to represent an animal's position in its environment, electroencephalogram data, or average calcium fluorescence as a function of time. Two objects were designed to represent groups of *Ts* and *Tsd*, namely *TsGroup* and *TsdFrame*. The main difference between the two objects is that *TsdFrame* has common timestamps for all the data (and therefore, all data have the same number of samples). *TsGroup* is more generic as each element has its own timestamps. These objects are typically used for ensembles of simultaneously recorded spike trains (*TsGroup*) or simultaneously acquired calcium fluorescence (*TsdFrame*). They are useful when operations need to be performed on a common time basis, for example binning multiple spike trains. Note however that they can be used for many other data types, for example the position of the animal (*TsdFrame*). Last, *IntervalSet* objects represent time epochs, for example the start and end times of intervals in which the animal is running.

Pynapple is built with objects from the Pandas library (*McKinney, 2011*). As such, Pynapple objects inherit the long-term consistency of the code and the computational flexibility from this widely used package. Specifically, a *Tsd* object is an extension of (or 'inherits' in object-orienting programming) Pandas *Series* object and *TsdFrame* of Pandas *DataFrame* object. A *TsGroup* is a child of *UserDict*, a built-in python object for inheriting dictionaries. Finally, *IntervalSet* inherits Pandas *DataFrame*. Time-stamps are by default in units of seconds but can be readily converted to other time units using the `as_units` method in any object.

Pynapple objects have a limited number of core methods (*Figure 2A*), which form the foundation of further operations. These operations provide a general framework by which users can manipulate the timestamps and their corresponding values as needed for analysis. For example, the time series objects have built-in methods: `value_from`, which gets the value from one time series object at the (closest) timestamps from another; `restrict`, which 'restricts' a time series object, extracting only the data contained within a set of time intervals defined by an *IntervalSet* object; `count`, which counts the number of timestamps from a time series object in windows of a given bin size; `threshold`, which applies a threshold to the data within a *Ts* or *Tsd* object and returns a *Tsd* containing the data above or below the threshold. All operations can be restricted to a given epoch, specified by an *IntervalSet*.

Furthermore, all objects have a `time_support` property, which keeps track of the time interval over which the data is valid. The time support is an *IntervalSet* object that is attached by default to *Ts*, *Tsd*, *TsdFrame,* and *TsGroup* objects. This is a crucial property as, otherwise, it is impossible to know whether periods without data correspond to an epoch during which the underlying event was not observed or because this period has previously been excluded by a restrict method.

In addition to the ability to restrict methods of time series objects, the *IntervalSet* object has methods for logical operations on combinations of *IntervalSets*, all returning other *IntervalSets* (*Figure 2b*): `intersect`, which returns the set intersection of two *IntervalSet* objects; `union`, which returns the set union of two *IntervalSets*; `set_diff`, which returns the set difference of two

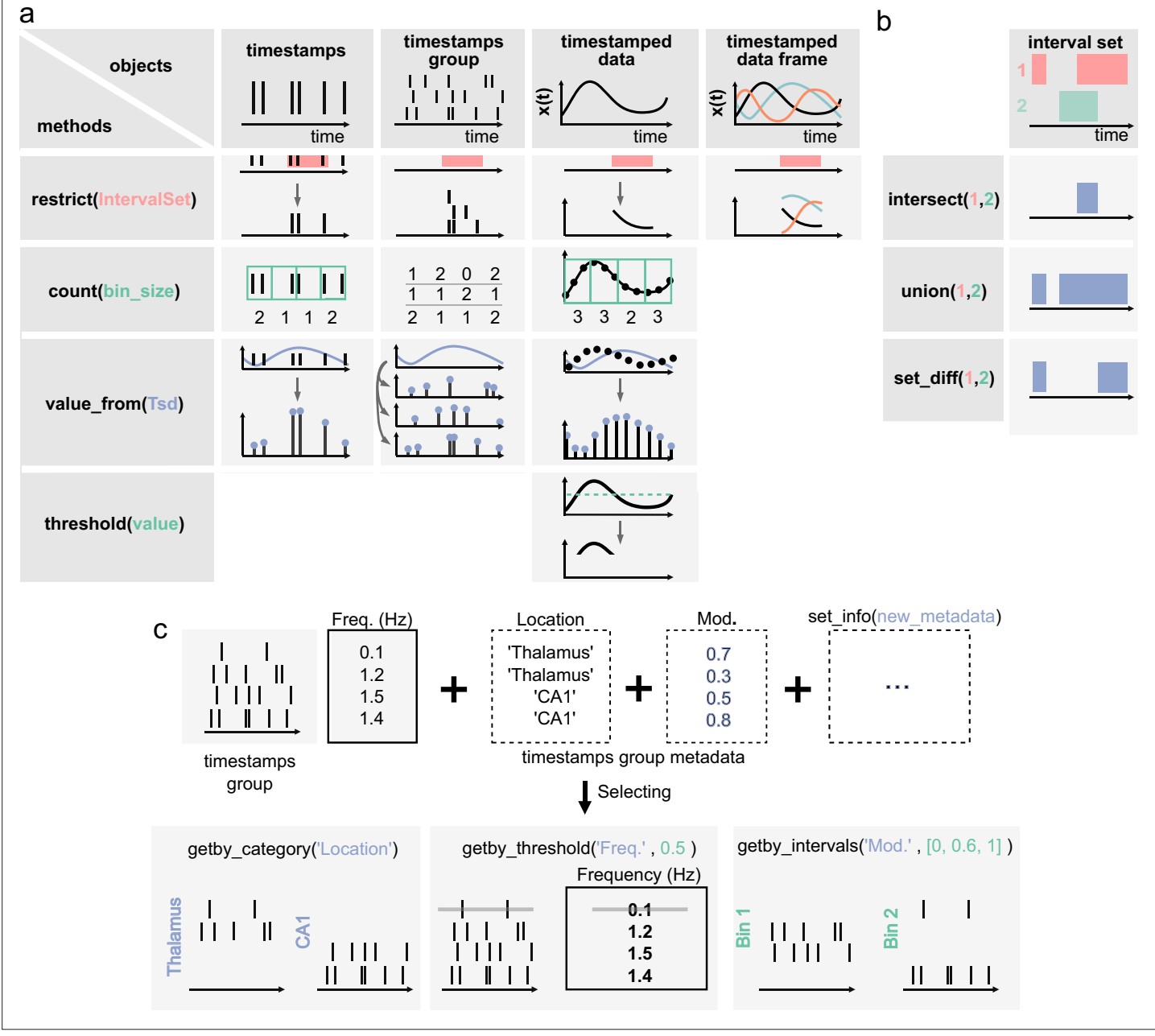

**Figure 2.** Core methods of the Pynapple objects. (**a**) Methods of timestamps (*Ts*) and timestamped data (*Tsd*) objects. The same methods can be called for different objects, leading to qualitatively similar results. For example, object.restrict(intervalset) returns an object now defined on the intersection of its original time support and the input *IntervalSet*. Objects can be any of the timestamps and timestamped data objects. These methods can be called with only one argument, as shown here, since the default parameters are typically the same for most analyses. Yet the methods include additional arguments for more specific operations. (**b**) Logical operations on pairs of *IntervalSet* objects to compute (from top to bottom) the intersection, union, and difference between epochs. These operations are commonly used to analyze data during specific epochs in a combinatorial manner, such as 'exploration period AND running speed is above 5 cm/s NOT left arm'. (**c**) Methods of *TsGroup* objects. Each timestamp is associated by default with its occurrence rate. Additional custom metadata such as recording location can be added. These metadata can then be used to select and filter timestamps using getby_category for discrete labels, getby_threshold, or getby_intervals for numerical values.

IntervalSet; `drop_short_intervals`, `drop_long_intervals`, which eliminate interval subsets that are shorter or longer than a desired duration; and `merge_close_intervals`, which merge intervals that are closer in time than a given duration.

Many experiments in neuroscience are based on trials, each associated with different conditions. *IntervalSets* are perfectly suited for this, as one *IntervalSet* can represent all start and end times

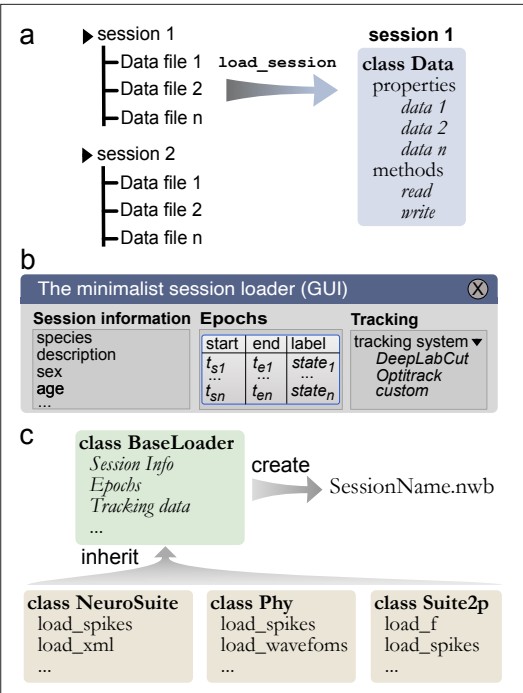

**Figure 3.** Built-in and customizable loading function for Pynapple. (**a**) Data is originally organized as separate files in a folder. A built-in or custom-made load_session function is called to load the data into a Data class. (**b**) Data can be loaded through a customizable graphical user interface (GUI) to enter all relevant information regarding the experiment, for example animal strain, among others. The main epochs of the recording (e.g., behavioral states, stimuli category, etc.) can be loaded from standard tabular data files (such as CSV). Behavioral tracking data extracted from various common systems and saved as a CSV file can also be loaded. (**c**) Pynapple offers various built-in loaders for commonly used data formats, as well as a template to easily design a customizable loader to adapt to any other format or specific task design.

of trials. The nature of each trial (e.g., left/right, correct/error) can be stored as a third column within the *IntervalSet* dataframe object. Thus, subsets of trials can be easily selected to restrict data of interest on the corresponding epochs. An alternative approach is to store different *Interval-Sets* for different types of trials.

In addition to the ability to apply any methods of the *Ts* object to its members, *TsGroup* has a set of methods to calculate and store metadata about the elements of the group (*Figure 2c*). For example, one can store and retrieve the anatomical structure from which a neuron was recorded, or the result from downstream analysis, perform operations on each element, and filter by various properties. These methods allow the user to, for example, calculate, store, and compare the properties of multiple neurons in a population. Additional methods for all objects are extensively documented in the documentation, and examples for usage are given in the tutorials.

While there are relatively few objects, and while each object has relatively few methods, they are the foundation of almost any analysis in systems neuroscience. However, if not implemented efficiently, they can be computationally intensive and if not implemented accurately, they are highly susceptible to user error. The implementation of core features in Pynapple addresses the concerns of efficiency and accuracy. Crucially, all units are indexed by seconds across the entire package, which limits the need for users to account for indexing and alignment between different streams of data at different sampling rates. For example, a user can simply use `spikes.value_from(position)` to get the animal's position at each spike time, rather than costly and error-prone routines in which a user needs to identify matching indices for the corresponding timestamps across arrays containing spikes and behavioral information.

Another common issue in data analysis is to analyze two time series that are not recorded at the same sampling rate. Once data are loaded in the same time base (i.e., the same time 0), they can keep their original sampling times. Using the function `value_from` from one object with the other object as argument will provide two time series with the same number of samples and the same sampling times, which will simplify further analyses. However, this means it is essential that all objects are loaded in the same time base for these methods to function correctly. Pynapple anticipates this by providing a customizable data loader, ensuring time bases are always loaded correctly.

## Importing data from common and custom pipelines
The proliferation of experimental methods has come with a proliferation of data formats, as well as the need to rapidly develop new formats that meet new experimental needs. Usually, these data formats are dependent on the software that was used to preprocess the raw data, making them difficult to load for further analysis. Additionally, an experimental setup can generate multiple streams of data that are saved within multiple files of various types. Thus, a universal toolbox should be able to load

popular data formats into a common framework and offer the user the ability to write functions to load their own data types.

To ease the process of loading and synchronizing data from various streams, Pynapple includes an I/O layer that allows the user to load multiple types of datasets and write them to a common format for further analysis and sharing. The primary way by which a user interacts with the I/O layer is an object that represents an experimental session, with the properties of the object being the various time series. This I/O object is created by calling the function `load_session`, which will load all data associated with that session (*Figure 3a*). For example, calling `load_session` for an *in vivo* electrophysiology recording would return an object called data, which will have properties `data.spikes`, `data.position`, and `data.epochs` which respectively store a *TsGroup,* containing the spike times, a *TsdFrame* containing the position of the animal, and an *IntervalSet* containing the times when the animal is on the track. With this object-oriented I/O method, the user can interact with the various data streams associated with a given experimental session and load multiple sessions at once without the risk of mixing data as each time series is attached to only one I/O object.

Data synchronization is the crux of any analysis pipeline. The `load_session` function is thus a crucial step in using the package. For unsupported data types, it is the responsibility of the users to design the preprocessing scripts that align the data streams in the same absolute time base. The data loading and synchronizing functions already included in the package for supported data types is a good starting point for any user writing a custom loading function (details of this process are provided later).

While data types are usually specific to a recording modality (i.e., calcium imaging and electrophysiology), there are several pieces of metadata that are common to many experiments, such as the strain of the animal, age, sex, and name of the experimenter. When loading a session for the first time, the I/O process starts with a graphical user interface (GUI) in which the user can quickly and easily input the general information as well as any session epoch and behavioral tracking data (*Figure 3b*). This information is saved in a *BaseLoader* class.

General session information is common across experimental sessions, however specialized data streams are usually specific to recording modalities. To cover the variety of preprocessing analysis pipelines currently used in systems neuroscience, the Pynapple I/O can load data formats from popular preprocessing pipelines (e.g., CNMF-E, Phy, NeuroSuite, or Suite2P). This is implemented via a set of specialized object subclasses of the BaseLoader class, avoiding the need to redefine I/O operations in each subclass. This is a core aspect of object-oriented programming, and it means that these specialized I/O classes have all the methods and properties of the parent *BaseLoader* objects. This ensures compatibility across various loading functions. However, once generated, these specialized I/O classes are unique and independent from each other, ensuring long-term backward compatibility. For instance, if the spike sorting tool Phy changes its output in the future, this would not affect the 'Neurosuite' IO class as they are independent of each other. This allows each tool to be updated or modified independently, without requiring changes to the other tool or the overall data format.

Like the *BaseLoader* class, a specialized GUI for electrophysiology and calcium imaging is provided, with relevant metadata fields, for example electrode position in electrophysiology and type of fluorescence indicator in calcium imaging (*Figure 3b*).

To avoid repeating the process of inputting session information and synchronization of multiple data streams, Pynapple saves all synchronized data into a unique file and can accommodate a wide range of neuroscientific data types. Recently, Neurodata Without Borders (NWB) (*Teeters et al., 2015*; *Rübel et al., 2022*) has emerged as a flexible data format used for public data sharing and large databases such as those collected by the Allen Institute. Thus, we chose to use the NWB format for fast and universal data loading and saving with Pynapple. The *BaseLoader* is responsible for initializing the NWB file within the session folder (i.e., it creates a new NWB file if none is present) (*Figure 3c*). Converting user's data to NWB format encourages standardization and can facilitate sharing both data and analysis pipelines written with Pynapple.

Many other preprocessing pipelines exist and can often be unique to a lab or even to an individual project. To accommodate present and future needs for these specific pipelines, the documentation of Pynapple provides an easy-to-follow recipe for creating a custom I/O class that inherits the *BaseLoader* and can interact with a pre-existing NWB file. There are multiple benefits of the inheritance approach of data loading classes within the I/O layer of Pynapple. First, future development of new

I/O classes will not affect the core and processing layers of Pynapple. This ensures long-term stability of the package. Second, users can develop their own custom I/O using available template classes. Pynapple already includes several of such templates and we expect this collection to grow in the future. Third, users can still use Pynapple without using the I/O layer of Pynapple. Last, in order to apply previous analyses, or analyses developed in another lab, to new data or data types a user only needs to develop a new I/O class for their data. This will import the data to the common Pynapple core from which the same analysis pipeline can be used.

## Foundational data processing

The basic methods that manipulate the core objects in Pynapple allow users to perform common, but powerful, neuroscience analyses (*Figure 2*). These analyses are easy to use because they describe the relationships between time series objects, while requiring the fewest number of parameters to be set by the user. This minimizes complexity, while maximizing generalizability. The operations in Pynapple can recreate neuroscience analyses from a broad number of subdisciplines. These analyses form the foundation of neuroscience data analysis in Pynapple. To illustrate the versatility of Pynapple and how it can be used, we reanalyzed five openly available datasets.

The first foundational analysis is computing neural tuning curves. Tuning curves relate specific stimuli to the firing rate of neurons. To this end, Pynapple computes the firing rate of a neuron (or any other timestamped data) during each epoch in an *IntervalSet* object, for example for discrete conditions such as 'ON/OFF'' stimuli. Tuning curves can also be computed with respect to a continuous feature. Once computed, Pynapple is able to use tuning curves from a population of neurons to decode stimuli using a Bayesian decoder (*Zhang et al., 1998*; *Brown et al., 1998*; *Figure 4a*).

The second foundational analysis is computing auto- and cross-correlograms of event data. In the most abstract sense, these correlograms show the relationship between previous and future events and a current event at time 0. In Pynapple, cross-correlograms can be generated for any two series of events by computing the event rate for each time bin of a target time series relative to each event of a reference time series. Commonly, this is used to examine the likelihood of an action potential in a neuron relating to a previous or future action potential in the same neuron (auto-correlogram) or in another neuron (cross-correlogram) (*Figure 4b*). However, Pynapple does not limit this function to spiking data and correlograms may be performed on any event-based data.

The third and final foundational analysis is peri-event alignment. This involves aligning a specified window from *Ts/Tsd/TsGroup* data to a specific *Ts*, known as 'TimeStamp Reference''. This allows users to align data to specific points in time, and measure changes in rates around this specified time point (*Figure 4c*). One example where this function is useful is aligning neuronal spikes to specific stimuli, such as optogenetic illumination, presentation of a tone, or electrical stimulation.

Some of the analyses presented so far are designed for spikes (and discrete events in general) and cannot be applied for continuous traces such as calcium imaging data. Pynapple includes specialized functions that can compute the tuning of a continuous value with respect to a feature, as shown for the modulation of fluorescence in calcium imaging with respect to the speed of the animal (*Figure 5a*) or of the position of a vertical bar on a screen in the fly's ellipsoid body (*Figure 5b*).

The examples shown in *Figures 4 and 5* show how these core analyses are useful for rapid data screening with just a few lines of code in a Jupyter notebook, for example. Overall, these foundational functions form the building blocks of most other analyses in systems neuroscience. Importantly, they are for the most part built-in and only depend on a few widely used external packages. This ensures that the package can be used in a near stand-alone fashion, without relying on packages that are at risk of not being maintained or of not being compatible in the near future. All other developments of analysis pipelines take place outside Pynapple, ensuring the core package is only updated rarely and remains lightweight.

## Pynacollada: a collaborative library for specialized and continuously updated data analyses

Pynapple is designed to be stable in the foreseeable future and its core functionality is not meant to be modified. However, actual data analysis usually requires more than the available core functions. This type of data analysis is 'fluid', constantly updated by new software developments and theoretical work. Furthermore, this kind of development is collaborative in nature and the supervision of such

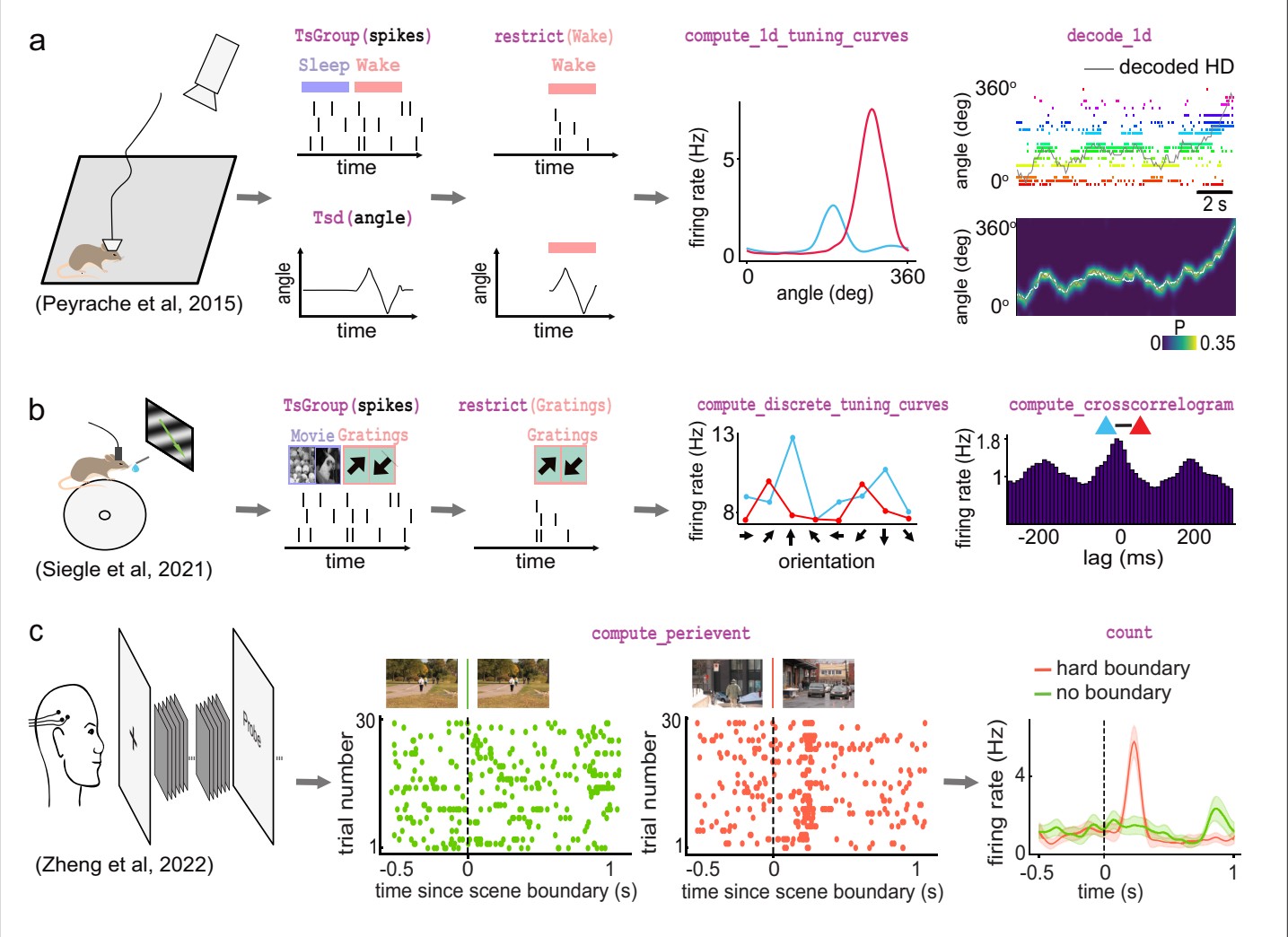

**Figure 4.** Examples of foundational analysis across various electrophysiological datasets using Pynapple. (**a**) Analysis of an ensemble of head-direction cells. From left to right: data were collected in a freely moving mouse randomly foraging for food; all data are restricted to the wake epoch (i.e., during exploration); the tuning curve of two neurons relative to the animal's head-direction; animal's head-direction is decoded from the neuronal ensemble. Data from *Peyrache et al., 2015a*; *Peyrache et al., 2015b*. (**b**) Analysis of V1 neurons during visual stimulation. From left to right: the mouse was recorded while being head-fixed and presented with drifting gratings; spikes, stimulation, and epochs are shown; example tuning curves of two V1 neurons, showing their firing rates for different grating orientations; example cross-correlation between two V1 neurons, showing an oscillatory co-modulation at about 5 Hz during visual stimulation. Data from *Siegle et al., 2021*. (**c**) Analysis of medial temporal lobe neurons in human epileptic subjects. From left to right: subjects, implanted with hybrid deep electrodes, were shown a series of short clips; raster plot of a single neuron around continuous movie shot trials (green) and hard boundary trials, which are transitions between two unrelated movies (orange); peri-event neuronal firing rate for both trial types. Data from *Zheng et al., 2022*. Images in panels b and c are from *Olmos and Kingdom, 2004*. The analysis code used to generate this figure can be found on the Pynapple Organization GitHub repository: https://github.com/pynapple-org/pynapple-paper-2023, swh:1:rev:2603975ce421a02a30b82a05a2c1bda810246f9d; (*Viejo, 2023a*).

projects is less sensitive than that of a stable package. To balance the needs for stability and flexibility, high-level functions were separated from Pynapple and included instead in Pynacollada: the Pynapple Collaborative repository hosted on GitHub.

Complex analyses are added to Pynacollada in the form of libraries. Each library developed for Pynacollada takes the form of a Jupyter notebook (or python scripts) which guides the user through the analysis step-by-step. As such, libraries built for Pynacollada should provide training, promote good practice in programming, and allow users to easily adapt code to their own project. Examples of complex analyses currently handled by Pynacollada are outlined below (*Figure 6*).

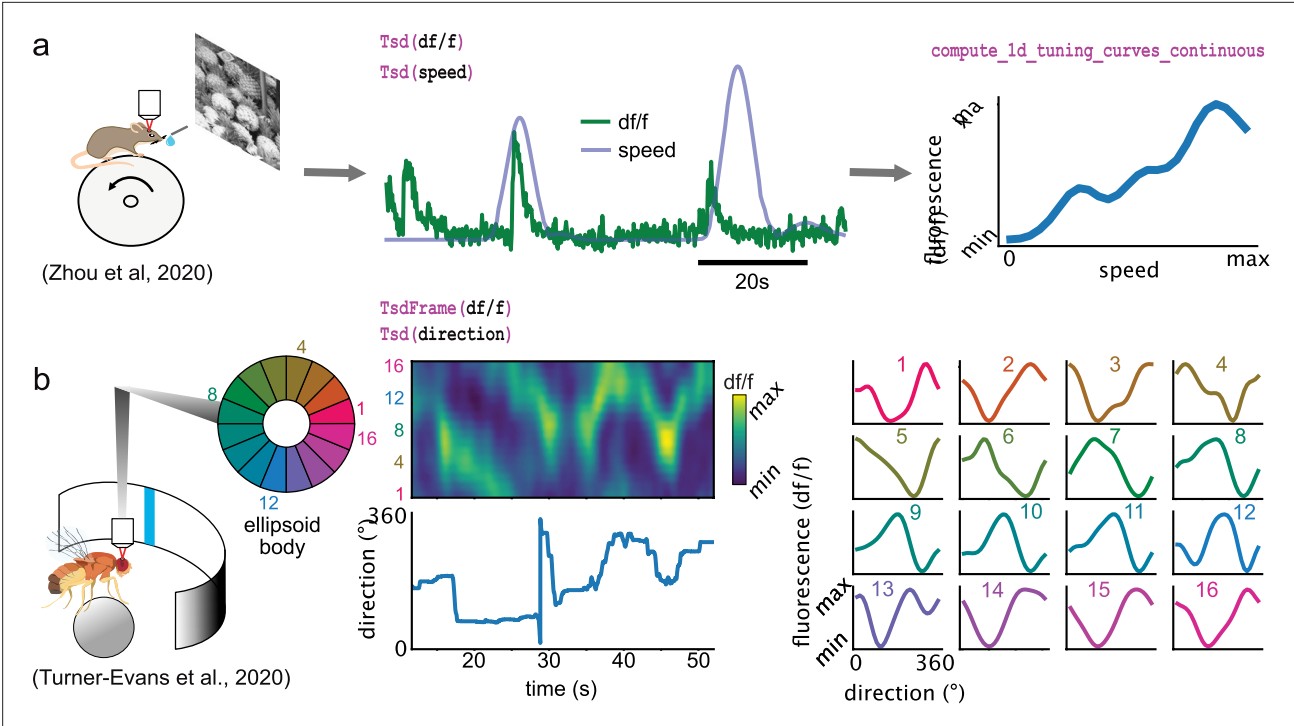

**Figure 5.** Examples of foundational analysis across various calcium imaging datasets using Pynapple. (**a**) Analysis of a V1 neuron during visual stimulation. From left to right: the mouse was recorded while being head-fixed on a running wheel and presented with natural scene movies; fluorescence traces from a preprocessed region of interest and running speed are loaded; continuous tuning curve is directly obtained from fluorescence and speed. Data from **Zhou et al., 2020**. Image is from **Olmos and Kingdom, 2004**. (**b**) Analysis of neuronal activity in the fly central complex. From left to right: a *Drosophila melanogaster* is tethered to a calcium imaging setup while the position of a vertical bar is in closed loop with the fly's movements on a ball; calcium activity in the ellipsoid body is divided into 16 wedges; example fluorescence trace and direction of the fly. Tuning curves are obtained as in (**a**), with the direction as feature. Data from **Turner-Evans et al., 2020**.

Recent advances in the application of manifold theory to neural data analysis have allowed neuroscientists to project high-dimensional data into three or fewer dimensions (**Chaudhuri et al., 2019**; **Viejo and Peyrache, 2020**; **Gardner et al., 2022**). The structure of these projections reflects the structure of these higher-dimensional processes, allowing us to infer the information encoded by the population. The Pynacollada 'neural_manifold' library contains a Jupyter notebook that provides a step-by-step process for recreating a ring manifold using spiking data recorded from a population of head-direction neurons (**Figure 6**). This code can be adapted by the end-user for analysis of their own data by simply importing their own data and refactoring the parameters to suit their needs.

A second complex analysis handled by Pynacollada is sharp wave-ripple (SWR) detection. Detecting oscillatory events is a routine procedure in electrophysiology, yet usually depends on many arbitrary choices of parameters. In this case, the Jupyter notebook showcases an example of detecting SWRs, a well-characterized oscillation of the hippocampus (**Figure 6**).

In addition, Pynacollada currently includes libraries for spike waveform processing, EEG analysis, and video tracking, among others. We invite the community to contribute to this repository by improving current libraries or upload new ones. For new libraries, only rapid screening and tests will be performed, but the code will not go through the kind of validation that is in place for Pynapple as an external library will never affect the functioning of the core package. The documentation describes what is expected in each library to simplify readability, sharing, and maintenance and, overall, how libraries should conform to Pynacollada standards. We hope this will be broadly adopted by the community, allowing researchers across labs to easily share their code.

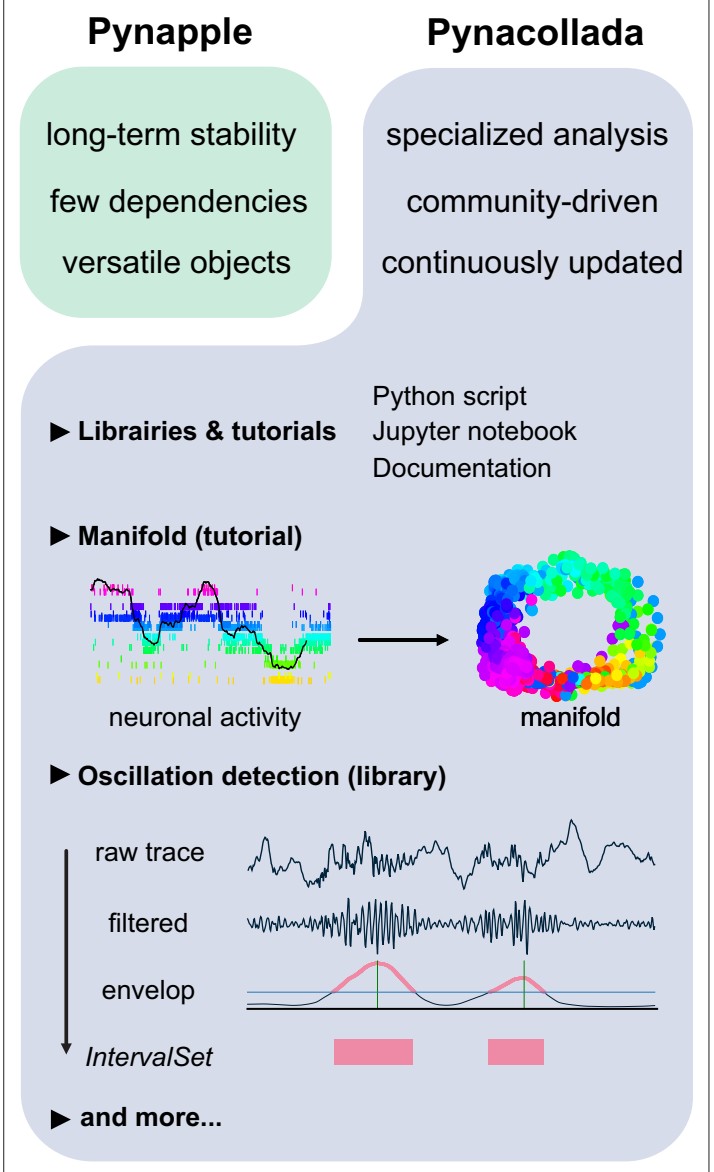

**Figure 6.** The Pynapple collaborative data analysis repository (Pynacollada) environment. Unlike Pynapple, which is designed for long-term stability, Pynacollada is a repository of project-oriented libraries. This way, the community can collaborate on constantly evolving data analysis code without affecting the functionality of the core pipeline. Each project should include a script that can be called for specific functions and/or Jupyter notebooks to showcase the use of the code, as well as proper documentation. Pynacollada already includes several libraries and/or tutorials, including but not limited to: (1) a tutorial on manifold analysis, covering how to project neuronal data on low-dimensional subspace using various machine learning techniques; (2) a library for oscillation detection in local field potentials, which takes raw broadband traces as inputs and outputs *IntervalSet* objects corresponding to the start and end times of oscillation bouts.

## Discussion

Here, we introduced Pynapple, a lightweight and open-source python toolbox for neural data analysis. The goal of this package is to offer a versatile set of tools to study typical neurophysiological and behavioral data, specifically time series (e.g., spike times, behavioral events, and continuous time series) and time intervals (e.g., trials and brain states). It also provides users with generic functions for neuroscience analyses such as tuning curves and cross-correlograms. Finally, Pynapple was designed to rely on a minimum number of dependencies, which are themselves very common and thus highly stable. As such, accessibility is the guiding axiom of Pynapple.

The path from data collection to reliable results involves a number of critical steps: exploratory data analysis, development of an analysis pipeline that can involve custom-made developed processing steps, and ideally the use of that pipeline and others to replicate the results. Pynapple provides a platform for these steps.

The design of Pynapple is centered around the manipulation of simple, abstract objects that are common to most neurophysiological and behavioral datasets. The core of Pynapple is built around five objects: timestamps (*Ts*) and group of timestamps (*TsGroup*), time series data (*Tsd*) and ensemble of co-registered Tsd (*TsdFrame*), as well as *IntervalSets*. These objects can be manipulated with properties that are, in most cases, common to all objects. Building around these fundamental objects and properties means Pynapple is highly flexible and able to handle most neurophysiological and behavioral datasets, making it accessible to most systems neuroscientists.

Pynapple was developed to be lightweight, stable, and simple. As simplicity does not necessarily imply backward compatibility (i.e., long-term stability of the code), Pynapple main objects and their properties will remain the same for the foreseeable future, even if the code in the backend may eventually change (e.g., not relying on Pandas in future versions). The small number of external dependencies also decreases the need to adapt the code to new versions of external packages. This approach favors long-term backward compatibility.

Data in neuroscience vary widely in their structure, size, and need for preprocessing. Pynapple is built around the idea that raw data have already been preprocessed (e.g., spike sorting and detection of ROIs). According to the FAIR principles, preprocessed data should interoperate across different analysis pipelines. Pynapple makes this interoperability possible as, once the data are loaded in the Pynapple framework, the same code can be used to analyze different datasets. Specifically, to simplify analysis for users, Pynapple offers simple wrappers for loading data with popular preprocessing pipelines. However, to be fully accessible, it is not sufficient for a package's core operations to be able to process all data types in theory. Data produced in neuroscience has a wide variety of file types, which are often only loaded by specific analysis software. Data is also largely experiment-specific. To unify these disparate file types and configurations, Pynapple's data loader is customizable. In addition to being able to load current popular data formats, this customizable data loader means emerging file formats may continue to be loaded in the future, without significant overhauls to the main package. This offers Pynapple long-term stability and means that Pynapple will continue to remain accessible in the foreseeable future. To note, Pynapple can be used without the I/O layer and independent of NWB for generic, on-the-fly analysis of data.

In further pursuit of accessibility, from these simple objects and properties, Pynapple has several built-in, foundational analyses that are common across the field of systems neuroscience. These foundational analyses include computing neural tuning curves, computing auto/cross-correlograms, peri-event alignment, and performing Bayesian decoding. From these foundational analyses, higher order analyses can be developed. However, these higher-order analyses are more prone to customization, thereby making them relatively more flexible. As such, higher-order analyses are stored in the collaborative repository known as Pynacollada. This keeps the core Pynapple package stable, while allowing the user to integrate new advances in neurophysiological and behavioral analysis into their workflow.

Other software provide programming environments which deal with common neuroscientific data and an interface between stored data and analytical methods (*Garcia et al., 2014*). However, one problem that arises from this structure is that objects and data structures are rigidly defined, leading to a lack of versatility for new types of data or task design. In contrast, Pynapple offers a more flexible working environment and will remain accessible even as user requirements change.

While Pynapple expands accessibility to data analysis, it has some limitations inherent to its design. The first issue is that Pynapple is currently only available through Python. Thus, some transition is required for those primarily trained in other programming languages commonly used in neuroscience, including MATLAB and Julia. The design of the package around objects is a strength in many regards but could represent a challenge for users who are not accustomed to this programming approach. We have addressed this concern by providing users with detailed documentation, which includes a broad variety of examples. We will also keep on providing training opportunities for all future users. Last, Python code may run slower than similar code written in other languages. Pynapple is based on Pandas, whose methods are already highly optimized. Yet, current development is underway to

improve computation speed and these developments are transparent for the users as they won't change the organization of the package.

Soon, Pynapple will be part of an entire suite of plugin libraries that we are developing to further enhance Pynapple. To keep Pynapple robust and stable, we will develop these plugins as standalone packages. These external packages will include an automated datalogger for recapitulating analyses, an online visualizer for Pynapple objects, and a package for parallel computing in Pynapple. This will address the speed issue inherent to code written in Python by allowing multiple analyses to be performed simultaneously. These packages will begin to address the limitations of Pynapple we described previously, enhancing the long-term stability of Pynapple, and streamlining accessibility for its users.

- Pynapple: https://github.com/pynapple-org/pynapple, copy archived at *Viejo, 2023b*.
- Pynacollada: https://github.com/PeyracheLab/pynacollada, copy archived at *Viejo, 2023c*.
- Code to generate *Figures 4 and 5*: https://github.com/pynapple-org/pynapple-paper-2023, copy archived at *Viejo, 2023a*.

## Code availability

## Acknowledgements

This work was supported by a Canadian Research Chair in Systems Neuroscience, CIHR Project Grant 155957 and 180330, NSERC Discovery Grant RGPIN-2018-04600, the Canada-Israel Health Research Initiative, jointly funded by the Canadian Institutes of Health Research, the Israel Science Foundation, the International Development Research Centre, Canada and the Azrieli Foundation 108877-001, and the Tanenbaum Open Science Institute (AP).

# Additional information

### Competing interests

Adrien Peyrache: Reviewing editor, *eLife*. The other authors declare that no competing interests exist.

### Funding

| Funder | Grant reference number | Author |
| --- | --- | --- |
| Canadian Institutes of Health Research | 155957 | Adrien Peyrache |
| Canadian Institutes of Health Research | 180330 | Adrien Peyrache |
| Natural Sciences and Engineering Research Council of Canada | RGPIN-2018-04600 | Adrien Peyrache |
| International Development Research Centre | 108877-001 | Adrien Peyrache |
| Tanenbaum Open Science Institute | | Adrien Peyrache |

The funders had no role in study design, data collection and interpretation, or the decision to submit the work for publication.

### Author contributions

Guillaume Viejo, Conceptualization, Resources, Data curation, Software, Formal analysis, Validation, Investigation, Visualization, Methodology, Writing – original draft, Writing – review and editing; Daniel Levenstein, Conceptualization, Software, Validation, Methodology, Writing – original draft, Writing – review and editing; Sofia Skromne Carrasco, Dhruv Mehrotra, Sara Mahallati, Software, Visualization, Writing – original draft, Writing – review and editing; Gilberto R Vite, Software, Visualization, Writing – review and editing; Henry Denny, Visualization, Writing – original draft, Writing – review and editing;

Lucas Sjulson, Conceptualization, Writing – review and editing; Francesco P Battaglia, Conceptualization, Software, Writing – review and editing; Adrien Peyrache, Conceptualization, Resources, Supervision, Funding acquisition, Validation, Visualization, Methodology, Writing – original draft, Writing – review and editing

### Author ORCIDs
Guillaume Viejo (iD) http://orcid.org/0000-0002-2450-7397
Daniel Levenstein (iD) http://orcid.org/0000-0002-5507-9145
Dhruv Mehrotra (iD) http://orcid.org/0000-0001-5813-3218
Adrien Peyrache (iD) http://orcid.org/0000-0001-9708-309X

Reviewer #1 (Public Review): https://doi.org/10.7554/eLife.85786.3.sa1
Reviewer #2 (Public Review): https://doi.org/10.7554/eLife.85786.3.sa2
Author Response https://doi.org/10.7554/eLife.85786.3.sa3

---

# Additional files

### Supplementary files
• MDAR checklist

### Data availability
All data used in this manuscript are publicly available and were previously published.

The following previously published datasets were used:

| Author(s) | Year | Dataset title | Dataset URL | Database and Identifier |
|---|---|---|---|---|
| Peyrache A, Petersen PC, Buzsaki G | 2015 | Extracellular recordings from multi-site silicon probes in the anterior thalamus and subicular formation of freely moving mice | http://dx.doi.org/10.6080/K0G15XS1 | Collaborative Research in Computational Neuroscience, 10.6080/K0G15XS1 |

*Continued on next page*

*Continued*

| Author(s) | Year | Dataset title | Dataset URL | Database and Identifier |
|---|---|---|---|---|
| Siegle JH, Jia X, Durand S, Gale S, Bennett C, Graddis N, Heller G, Ramirez TK, Choi H, Luviano JA, Groblewski PA, Ahmed R, Arkhipov A, Bernard A, Billeh YN, Brown D, Buice MA, Cain N, Caldejon S, Casal L, Cho A, Chvilicek M, Cox TC, Dai K, Denman DJ, de Vries SEJ, Dietzman R, Esposito L, Farrell C, Feng D, Galbraith J, Garrett M, Hancock N, Harris JA, Howard R, Hu B, Hytnen R, Iyer R, Jessett E, Johnson K, Kato I, Kiggins J, Lambert S, Lecoq J, Ledochowitsch P, Lee JH, Zeng H, Naylor S, Phillips JW, Reid C, Mihalas S, Olsen SR, Koch C, Leon A, Li Y, Liang E, Long F, Mace K, Melchior J, Millman D, Mollenkopf T, Nayan C, Ng L, Ngo K, Nguyen T, Nicovich PR, North K, Ocker GK, Ollerenshaw D, Oliver M, Pachitariu M, Reding M, Reid D, Robertson M, Ronellenfitch K, Seid S, Slaughterbeck C, Stoecklin M, Sullivan D, Sutton B, Swapp J, Thompson C, Turner K, Wakeman W, Whitesell JD, Williams D, Williford A, Young R | 2021 | Survey of spiking in the mouse visual system reveals functional hierarchy | https://gui.dandiarchive.org/#/dandiset/000021 | Dandiset, 000021 |
| Turner-Evans D, Jensen KT, Ali S, Paterson T, Sheridan A, Ray RP, Wolff T, Lauritzen S, Rubin GM, Bock DD, Jayaraman V | 2020 | The Neuroanatomical Ultrastructure and Function of a Biological Ring Attractor | https://janelia.figshare.com/articles/dataset/OneColor_zip/12490373 | Allen, 12490373 |
| Schjetnan A, Yebra M, Gomes BA, Mosher CP, Kalia SK, Valiante TA, Mamelak AN, Kreiman G, Rutishauser U, Zheng J | 2022 | Data for: Neurons detect cognitive boundaries to structure episodic memories in humans (Zheng et al., 2022, Nat Neuro in press) | https://doi.org/10.48324/dandi.000207/0.220216.0323 | Dandiset, 10.48324/dandi.000207/0.220216.0323 |

*Continued on next page*

*Continued*

| Author(s) | Year | Dataset title | Dataset URL | Database and Identifier |
|---|---|---|---|---|
| Zhou P, Reimer J, Zhou D, Pasarkar A, Kinsella I, Froudarakis E, Yatsenko DV, Fahey PG, Bodor A, Buchanan J, Bumbarger D, Mahalingam G, Torres R, Dorkenwald S, Ih D, Lee K, Lu R, Macrina T, Wu J, Costa N, Reid C, Tolias AS, Paninski L | 2020 | EASE: EM-Assisted Source Extraction from calcium imaging data | https://www.microns-explorer.org/cortical-mm3#f-data | Allen, cortical-mm3#f-data |

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
