## [Editor Report · eLife assessment]

This paper introduces the python software package Pynapple and a separate package of more advanced routines (Pynacollada) to the Neuroscience/Neural Engineering community. Pynapple provides a set of data objects and methods that have the potential to simplify data analysis for neural and behavioral data types. This represents a **valuable** contribution to the field. With more examples and as a live coding notebook, the evidence was judged to be **compelling**.

---

## [Referee Report · Reviewer #1 (Public Review)]

A typical path from preprocessed data to findings in systems neuroscience often includes set of analyses that often share common components. For example, an investigator might want to generate plots that relating one time series (e.g., a set of spike times) to another (measurements of a behavioral parameter such as pupil diameter or running speed). In most cases, each individual scientist writes their own code to carry out these analyses, and thus the same basic analysis is coded repeatedly. This is problematic for several reasons, including the inefficiency of different people writing the same code over and over again.

This paper presents Pynapple, a python package that aims to address those problems.

Strengths:

The authors have identified a key need in the community - well written analysis routines that carry out a core set of functions and can import data from multiple formats. In addition, they recognized that there are some common elements of many analyses, particularly those involving timeseries, and their object-oriented architecture takes advantage of those commonalities to simplify the overall analysis process.

The package is separated into a core set of applications and another with more advanced applications, with the goal of both providing a streamlined base for analyses and allowing for implementations/inclusion of more experimental approaches.

Weaknesses:

The revised version of the paper does a very good job of addressing previous concerns. It would be slightly more accurate in the Highlights section to say "A lightweight and standalone package facilitating long-term backward compatibility" but this is a very minor issue.

---

## [Referee Report · Reviewer #2 (Public Review)]

The manuscript by G. Viejo et al. describes a new open-source toolbox called Pynapple, for data analysis of electrophysiological recordings, calcium imaging, and behavioral data. It is an object-oriented python package, consisting of 5 main object types: timestamps (Ts), timestamped data (Tsd), TsGroup, TsdFrame, and IntervalSet. Each object has a set of built-in core methods and import tools for common data acquisition systems and pipelines.

Pynapple is a low-level package that uses NWB as a file format, and further allows for other more advanced toolsets to build upon it. One of these is called Pynacollada which is a toolset for data analysis of electrophysiological, imaging, and behavioral data.

Pynapple and Pynacollada have the potential to become very valuable and foundational tools for the analysis of neurophysiological data. NWB still has a steep learning curve and Pynapple offers a user-friendly toolset that can also serve as a wrapper for NWB.

---

## [Author Response]

The following is the authors’ response to the original reviews.

We would ﬁrst like to thank the reviewers and the editor for their insightful comments and suggestions. We are particularly glad to read that our so<ware package constitutes a set of “well-written analysis routines” which have “the potential to become very valuable and foundational tools for the analysis of neurophysiological data”. We have updated the manuscript to address their remarks where appropriate.

Additionally, we would like to stress that this kind of tools is in continual development. As such, the manuscript oﬀered a snapshot of the package at one point during this process, which in this case was several months ago at initial submission. Since then, several improvements were implemented. The manuscript has been further updated to reﬂect these more recent changes.

From the Reviewing Editor:The reviewers identiﬁed a number of fundamental weaknesses in the paper.1. For a paper demonstrating a toolbox, it seems that some example analyses showing the value of the approach (and potentially the advantage in simpliﬁcation, etc over previous or other approaches) are really important to demonstrate.

As noted by the ﬁrst reviewer, the online repository (i.e. GitHub page) conveys a better sense of the toolboxes’ contribution to the ﬁeld than the present manuscript. This is a fair remark but at the same time, it is unclear how to illustrate this in a journal article without dedicating a great deal of page space to presenting raw code, while online tools oﬀer an easier and clearer way to do this. As a work-around, our strategy was to illustrate some examples of data analysis in Figures 4&5 by comparing each illustrated processing step to the corresponding command line used by the Pynapple package. Each step requires a single line of code, meaning that one only needs to write three lines of code to decode a feature from population activity using a Bayesian decoder (Fig. 4a), compute a cross-correlograms of two neurons during speciﬁc stimulus presentation (Fig. 4b) or compute the average ﬁring rate of two neurons around a speciﬁc time of the experimental task (Fig. 4c). We believe that these visual aides make it unnecessary to add code in the main text of this manuscript. However, to aid reader understanding, we now provide clear references to online Jupyter notebooks which show how each ﬁgure was generated in ﬁgure legends as well as in the “Code Availability” section.

https://github.com/pynapple-org/pynapple-paper-2023

Furthermore, we have opted-in for the “Executable Research Articles” feature at eLife, which will make it possible to include live scripts and ﬁgures in the manuscript once it is accepted for publication. We do not know at this stage what it entails exactly, but we hope that Figures 4&5 will become live with this feature. The readers will have the possibility to see and edit the code directly within the online version of the manuscript.

1. The manuscript's claims about not having dependencies seem confusing.

We agree that this claim was somewhat unfounded. There are virtually no Python packages that do not have dependencies. Our intention was to say that the package had no dependencies outside the most common ones, which are Numpy, Scipy, and Pandas. Too many packages in the ﬁeld tend to have long list of dependencies making long-term back-compatibility quite challenging. By keeping depencies minimal, we hope to maximise the package’'s long term back-compatibility. We have rephrased this statement in the manuscript in the following sections:

Figure 1, legend.

“These methods depend only on a few, commonly used, external packages.”

Section Foundational data processing:“they are for the most part built-in and only depend on a few widely-used external packages. This ensures that the package can be used in a near stand-alone fashion, without relying on packages that are at risk of not being maintained or of not being compatible in the near future.”

1. Given its signiﬁcant relevance, it seems important to cite the FMATool and describe connections between it (or analyses based on it) and the presented work.

Indeed, although we had already cited other toolboxes (including a review covering the topic comprehensively), we should have included this one in the original manuscript. Unfortunately, to the best of our knowledge, this toolbox is not citable (there is no companion paper). We have added a reference to it in plain text.

1. Some discussion of integration between Pynapple and the rest of a full experimental data pipeline should be discussed with regard to reproducibility.

This is an interesting point, and the third paragraph of the discussion somewhat broached this issue. Pynapple was not originally designed to pre-process data. However, it can, in theory, load any type of data streams a<er the necessary pre-processing steps. Overall, modularity is a key aspect of the Pynapple framework, and this is also the case for the integration with data pre-processing pipelines, for example spike sorting in electrophysiology and detection of region of interest in calcium imaging. We do not think there should be an integrated solution to the problem but, instead, to make it possible that any piece of code can be used for data irrespective of their origin. This is why we focused on making data loading straightforward and easy to adapt to any particular situation.To expand on this point and make it clear that Pynapple is not meant to pre-process data but can, in theory, load any type of data streams a<er the necessary pre-processing steps, we have added the following sentences to the aforementioned paragraph:

“Data in neuroscience vary widely in their structure, size, and need for pre-processing. Pynapple is built around the idea that raw data have already been pre-processed (for example, spike sorting and detection of ROIs).”

1. Relatedly, a description of how data are stored a<er processing (i.e., how precisely are processed data stored in NWB format).

We agree that this is a critical issue. NWB is not necessarily the best option as it is not possible to overwrite in a NWB ﬁle. This would require the creation of a new NWB ﬁle each time, which is computationally expensive and time consuming. It also further increases the odds of writing error. Theoretically, users who needs to store intermediate results in a ﬂexible way could use any methods they prefer, writing their own data ﬁles and wrappers to reload these data into Pynapple objects. Indeed, it is not easy to properly store data in an object-speciﬁc manner. This is a long-standing issue and one we are currently working to resolve.

To do so, we are developing I/O methods for each Pynapple core objects. We aim to provide an output format that is simple to read and backward compatible in future Pynapple releases. This feature will be available in the coming weeks. To note, while NWB may not be the central data format of Pynapple in future releases, it has become a central node in the neuroscience ecosystem of so<ware. Therefore, we aim to facilitate the interaction of users with reading and writing for this format by developing a set of simple standalone functions.

**Reviewer #1 (Public Review):**
A typical path from preprocessed data to ﬁndings in systems neuroscience o<en includes a set of analyses that o<en share common components. For example, an investigator might want to generate plots that relate one time series (e.g., a set of spike times) to another (measurements of a behavioral parameter such as pupil diameter or running speed). In most cases, each individual scientist writes their own code to carry out these analyses, and thus the same basic analysis is coded repeatedly. This is problematic for several reasons, including the waste of time, the potential for errors, and the greater diﬃculty inherent in sharing highly customized code.This paper presents Pynapple, a python package that aims to address those problems.Strengths:The authors have identiﬁed a key need in the community - well-written analysis routines that carry out a core set of functions and can import data from multiple formats. In addition, they recognized that there are some common elements of many analyses, particularly those involving timeseries, and their object- oriented architecture takes advantage of those commonalities to simplify the overall analysis process.The package is separated into a core set of applications and another with more advanced applications, with the goal of both providing a streamlined base for analyses and allowing for implementations/inclusion of more experimental approaches.Weaknesses:There are two main weaknesses of the paper in its present form.First, the claims relating to the value of the library in everyday use are not demonstrated clearly. There are no comparisons of, for example, the number of lines of code required to carry out a speciﬁc analysis with and without Pynapple or Pynacollada. Similarly, the paper does not give the reader a good sense of how analyses are carried out and how the object-oriented architecture provides a simpliﬁed user interaction experience. This contrasts with their GitHub page and associated notebooks which do a better job of showing the package in action.

As noted in the response to the Reviewing Editor and response to the reviewer’s recommendation to the authors below, we have now included links to Jupyter notebooks that highlight how panels of Figures 4 and 5 were generated (https://github.com/pynapple-org/pynapple-paper-2023). However, we believe that including more code in the manuscript than what is currently shown (I.e. abbreviated call to methods on top of panels in Figs 4&5) would decrease the readability of the manuscript.

Second, the paper makes several claims about the values of object-oriented programming and the overall design strategy that are not entirely accurate. For example, object-oriented programming does not inherently reduce coding errors, although it can be part of good so<ware engineering. Similarly, there is a claim that the design strategy "ensures stability" when it would be much more accurate to say that these strategies make it easier to maintain the stability of the code. And the authors state that the package has no dependencies, which is not true in the codebase. These and other claims are made without a clear deﬁnition of the properties that good scientiﬁc analysis so<ware should have (e.g., stability, extensibility, testing infrastructure, etc.).

Following thFMAe reviewer’s comment, we have rephrased and clariﬁed these claims. We provide detailed response to these remarks in the recommendations to authors below.

There is also a minor issue - these packages address an important need for high-level analysis tools but do not provide associated tools for preprocessing (e.g., spike sorting) or for creating reproducible pipelines for these analyses. This is entirely reasonable, in that no one package can be expected to do everything, but a bit deeper account of the process that takes raw data and produces scientiﬁc results would be helpful. In addition, some discussion of how this package could be combined with other tools (e.g., DataJoint, Code Ocean) would help provide context for where Pynapple and Pynacollada could ﬁt into a robust and reliable data analysis ecosystem.

We agree the better explaining how Pynapple is integrated within data preprocessing pipelines is essential. We have clariﬁed this aspect in the manuscript and provide more details below.

**Reviewer #1 (Recommendations For The Authors):**
Page 1TitleThe authors should note that the application name- "Pynapple" could be confused with something from Apple. Users may search for "Pyapple" as many python applications contain "py" like "Numpy". "Pyapple" indeed is a Python Apple that works with Apple products. They could consider "NeuroFrame", "NeuroSeries" or "NeuroPandas" to help users realize this is not an apple product.

We thank the referee for this interesting comment. However, we are not willing to make such change at this point. The community of users has been growing in the last year and it seems too late to change the name. To note, it is the ﬁrst time such comment is made to us and it does not seem that users and collaborators are confused with any Apple products.

AbstractThe authors mentioned that the Pynapple is "fully open source". It may be better to simply say it is "open source".

We agree, corrected.

Assuming the authors keep the name, it would be helpful if the full meaning of Pynapple - Python Neural Analysis Package was presented as early as possible.

Corrected in the abstract.

HighlightAn application being lightweight and standalone does not imply nor ensure backward compatibility. In general, it would be useful if the authors identiﬁed a set of desirable code characteristics, deﬁned them clearly in the introduction, and then describe their so<ware in terms of those characteristics.

Thank you for your comment. We agree that being lightweight and standalone does not necessarily imply backward compatibility. Our intention was to emphasize that Pynapple is designed to be as simple and ﬂexible as possible, with a focus on providing a consistent interface for users across diﬀerent versions. However, we understand that this may not be enough to ensure long-term stability, which is why we are committed to regular updates and maintenance to ensure that the code remains functional as the underlying code base (Python versions, etc.) changes.

Regarding your suggestion to identify a set of desirable code characteristics, we believe this is an excellent idea. In the introduction, we brieﬂy touch upon some of the core principles that guided our development of Pynapple: a lightweight, stable, and simple package. However, we acknowledge that providing a more detailed discussion of these characteristics and how they relate to the design of our so<ware would be useful for readers. We have added this paragraph in the discussion:

“Pynapple was developed to be lightweight, stable, and simple. As simplicity does not necessarily imply backward compatibility (i.e. long-term stability of the code), Pynapple main objects and their properties will remain the same for the foreseeable future, even if the code in the backend may eventually change (e.g. not relying on Pandas in future version). The small number of external dependencies also decrease the need to adapt the code to new versions of external packages. This approach favors long-term backward compatibility.”

Page 2The authors wrote -"Despite this rapid progress, data analysis o<en relies on custom-made, lab-speciﬁc code, which is susceptible to error and can be diﬃcult to compare across research groups."It would be helpful to add that custom-made, lab-speciﬁc code can lead to a violation of FAIR principles (https://en.wikipedia.org/wiki/FAIR_datadata). More generally, any package can have errors, so it would be helpful to explain any testing regiments or other approach the authors have taken to ensure that their code is error-free.

We understand the importance of the FAIR principles for data sharing. However, Pynapple was not designed to handle data through their pre-processing. The only aspect that is somehow covered by the FAIR principles is the interoperability, but again, it is a requirement for the data to interoperate with diﬀerent storage and analysis pipelines, not of the analysis framework itself. Unlike custom-made code, Pynapple will make interoperability easier, as, in theory, once the required data loaders are available, any analysis could be run on any dataset. We have added the following sentence to the discussion:

“Data in neuroscience vary widely in their structure, size, and need for pre-processing. Pynapple is built around the idea that raw data has already been pre-processed (for example, spike sorting and ROI detection). According to the FAIR principles, pre-processed data should interoperate across diﬀerent analysis pipelines. Pynapple makes this interoperability possible as, once the data are loaded in the Pynapple framework, the same code can be used to analyze diﬀerent datasets”

The authors wrote -"While several toolboxes are available to perform neuronal data analysis ti‚Äì11,2ti (see ref. 29 for review), most of these programs focus on producing high-level analysis from speciﬁed types of data and do not oﬀer the versatility required for rapidly-changing analytical methods and experimental methods."Here it would be helpful if the authors could give a more speciﬁc example or explain why this is problematic enough to be a concern. Users may not see a problem with high-level analysis or using speciﬁc data types.

Again, we apologize for not fully elaborating upon our goals here. Our intention was to point out that toolboxes o<en focus on one particular case of high-level analysis. In many cases, such packages lack low level analysis features or the ﬂexibility to derive new analysis pipelines quickly and eﬀortlessly. Users can decide to use low-level packages such as Pandas, but in that case, the learning curve can be steep for users with low, if any, computational background. The simplicity of Pynapple, and the set of examples and notebooks, make it possible for individuals who start coding to be quickly able to analyze their data.

As we do not want to be too speciﬁc at this point of the manuscript (second paragraph of the intro) and as we have clariﬁed many of the aspects of the toolbox in the new revised version, we have only added the following sentence to the paragraph:

“Users can decide to use low-level data manipulation packages such as Pandas, but in that case, the learning curve can be steep for users with low, if any, computational background.”

The authors wrote -"To meet these needs, a general toolbox for data analysis must be designed with a few principles in mind"Toolboxes based on many diﬀerent principles can solve problems. It is likely more accurate to say that the authors designed their toolbox with a particular set of principles in mind. A clear description of those principles (as mentioned in the comment above) would help the reader understand why the speciﬁc choices made are beneﬁcial.

We agree that these are not “universal” principles and clearly more the principles we had in mind when we designed the package. We have clariﬁed these principles and made clear that these are personal point of views.

We have rephrased the following paragraph:

“To meet these needs, we designed Pynapple, a general toolbox for data analysis in systems Neuroscience with a few principles in mind.“

The authors wrote -"The ﬁrst property of such a toolbox is that it should be object-oriented, organizing so<ware around data."What facts make this true? For example, React is a web development library. A common approach to using this library is to use Hooks (essentially a collection of functions). This is becoming more popular than the previous approach of using Components (a collection of classes). This is an example of how Object-oriented programming is not always the best solution. In some cases, for example, object- oriented coding can cause problems (e.g. it can be hard to ﬁnd the place where a given function is deﬁned and to ﬁgure out which version is being used given complex inheritance structures.)In general, key selling points of object-oriented programming are extension, inheritance, and encapsulation. If the authors want to retain this text (which would be entirely reasonable), it would be helpful if they explained clearly how an object-oriented approach enables these functions and why they are critical for this application in particular.

The referee makes a particularly important point. We are aware of the limits of OOP, especially when these objects become over-complex, and that the inheritance become unclear.

We have clariﬁed our goal here. We believe that in our case, OOP is powerful and, overall, is less error- prone that a collection of functions. The reasons are the following:

An object-oriented approach facilitates better interactions between objects. By encapsulating data and behavior within objects, object-oriented programming promotes clear and well-deﬁned interfaces between objects. This results in more structured and manageable code, as objects communicate with each other through these well-deﬁned interfaces. Such improved interactions lead to increased code reliability.

Inheritance, a key concept in object-oriented programming, allows for the inheritance of properties. One important example of how inheritance is crucial in the Pynapple framework is the time support of Pynapple objects. It determines the valid epoch on which the object is deﬁned. This property needs to be carried over during diﬀerent manipulations of the object. Without OOP, this property could easily be forgotten, resulting in erroneous conclusions for many types of analysis. The simplest case is the average rate of a TS object: the rate must be computed on the time support ( a property of TS objects), not the beginning to the end of the recording (or of a speciﬁc epoch, independent of the TS). Finally, it is easier to access and manipulate the meta information of a Pynapple object than without using objects.

The authors wrote -"drastically diminishing the odds of a coding error"This seems a bit strong here. Perhaps "reducing the odds" would be more accurate.

We agree. Now changed.

Page 3The authors wrote -". Another property of an eﬃcient toolbox is that as much data as possible should be captured by only a small number of objects This ensures that the same code can be used for various datasetsand eliminates the need of adapting the structure"It may be better to write something like - "Objects have a collection of preset variables/values that are well suited for general use and are very ﬂexible." Capturing "as much data as possible" may be confusing, because it's not the amount that this helps with but rather the variety.

We thank the referee for this remark. We have rephrased this sentence as follows:

“Another property of an eﬃcient toolbox is that a small number of objects could virtually represents all possible data streams in neuroscience, instead of objects made for speciﬁc physiological processes (e.g. spike trains).”

The authors wrote -"The properties listed above ensure the long-term stability of a toolbox, a crucial aspect for maintaining the code repository. Toolboxes built around these principles will be maximally ﬂexible and will have the most generalapplication"There are two issues with this statement. First, ensuring long-term stability is only possible with a long- term commitment of time and resources to ensure that that code remains functional as the underlying code base (python versions, etc.) changes. If that is something you are commisng to, it would be great to make that clear. If not, these statements need to be less ﬁrm.Second, it is not clear how these properties were arrived at in the ﬁrst place. There are things like the FAIR Principles which could provide an organizing framework, ideally when combined with good so<ware engineering practices, and if some more systematic discussion of these properties and their justiﬁcation could be added, it would help the ﬁeld think about this issue more clearly.

The referee makes a valid point that ensuring long-term stability requires a long-term commitment of time and resources to maintain the code as the underlying technology evolves. While we cannot make guarantees about the future of Pynapple, we believe that one of the best ways to ensure long-term stability is by fostering a strong community of users and contributors who can provide ongoing support and development. By promoting open-source collaboration and encouraging community involvement, we hope to create a sustainable ecosystem around Pynapple that can adapt to changes in technology and scientiﬁc practices over time. Ultimately, the longevity of any scientiﬁc tool depends on its adoption and use by the research community, and we hope that Pynapple can provide value to neuroscience researchers and continue to evolve and improve as the ﬁeld progresses.

It is noteworthy that the ﬁrst author, and main developer of the package, has now been hired as a data scientist at the Center for Computational Neuroscience, Flatiron Institute, to explicitly continue the development of the tool and build a community of users and contributors.

The authors wrote -"each with a limited number of methods..."This may give the impression that the functionality is limited, so rephrasing may be helpful.

Indeed! We have now rephrased this sentence:

“The core of Pynapple is ﬁve versatile timeseries objects, whose methods make it possible to intuitively manipulate and analyze the data.”

The authors wrote that object-oriented coding"limits the chances of coding error"This is not always the case, but if it is the case here, it would be helpful if the authors explain exactly how it helps to use object-oriented approaches for this package.

We agree with the referee that it is not always the case. As we explained above, we believe it is less error-prone that a collection of functions. Quite o<en, it also makes it easier to debug.We have changed this sentence with the following one:

“Because objects are designed to be self-contained and interact with each other through well-deﬁned methods, users are less likely to make errors when using them. This is because objects can enforce their own internal consistency, reducing the chances of data inconsistencies or unexpected behavior. Overall, OOP is a powerful tool for managing complexity and reducing errors in scientiﬁc programming.”

Fig 1In object-oriented programming, a class is a blueprint for the classes that inherit it. Instantiating thatclass creates an object. An object contains any or all of these - data, methods, and events. The ﬁgure could be improved if it maintained these organizational principles as ﬁgure properties.

We agree with the referee’s remark regarding the logic of objects instantiation but how this could be incorporated in Fig. 1 without making it too complex is unclear. Here, objects are instantiated from the ﬁrst to the second column. We have not provided details about the parent objects, as we believe these details are not important for reader comprehension. In its present form, the objects are inherited from Pandas objects, but it is possible that a future version is based on something else. For the users, this will be transparent as the toolbox is designed in such a way that only the methods that are speciﬁc to Pynapple are needed to do most computation, while only expert programmers may be interested in using Pandas functionalities.

The authors wrote that Pynapple does -"not depend on any external package"As mentioned above, this is not true. It depends on Numpy and likely other packages, and this should be explained. It is perfectly reasonable to say that it depends on only a few other packages.

As said above, we have now clariﬁed this claim.

Page 5.The authors wrote -"represent arrays of Ts and Tsd"For a knowledgeable reader's reference, it would be helpful to refer to these either as Numpy arrays (at least at ﬁrst when they are deﬁned) or as lists if they are native python objects.

Indeed, using the word “arrays” here could be confusing because of Numpy arrays. We have changed this term with “groups”.

The authors wrote -"Pynapple is built with objects from the Pandas library ... Pynapple objects inherit the computational stability and ﬂexibility"Here a deﬁnition of stability would be useful. Is it the case that by stability you mean "does not change o<en"? Or is some other meaning of stability implied?

Yes, this is exactly what we meant when referring to the stability of Pandas. We have added the following precision:

“As such, Pynapple objects inherit the long-term consistency of the code and the computational ﬂexibility computational stability and ﬂexibility from this widely used package.”

Page 6Fig 2In Fig 2 A and B, the illustrations are good. It would also be very helpful to use toy code examples to illustrate how Pynapple will be used to carry out on a sample analysis-problem so that potential users can see what would need to be done.

We appreciate the kind works. Regarding the toy code, this is what we tried to do in Fig. 4. Instead of including the code directly in the paper, which does not seem a modern way of doing this, we now refer to the online notebooks that reproduce all panels of Figure 4.

The authors wrote -"While these objects and methods are relatively few"In object-oriented programming, objects contain methods. If a method is not in an object, it is not technically a method but a function. It would be helpful if the authors made sure their terminology is accurate, perhaps by saying something like "While there are relatively few objects, and while each object has relatively few methods ... "

We agree with the referee, we have changed the sentence accordingly.

The authors wrote -"if not implemented correctly, they can be both computationally intensive and highly susceptible to user error"Here the authors are using "correctly" to refer to two things - "accuracy" - gesng the right answer, and "eﬃciency" - gesng to that answer with relatively less computation. It would be clearer if they split out those two concepts in the phrasing.

Indeed, we used the term to cover both aspects of the problem, leading to the two possible issues cited in the second part of the sentence. We have changed the sentence following the referee’s advice:

“While there are relatively few objects, and while each object has relatively few methods, they are the foundation of almost any analysis in systems neuroscience. However, if not implemented eﬃciently, they can be computationally intensive and if not implemented accurately, they are highly susceptible to user error.”

In the next sentence the authors wrote -"Pynapple addresses this concern."This statement would beneﬁt from just additional text explaining how the concern is addressed.

We thank the referee for the suggestion. We have changed the sentence to this one:“The implementation of core features in Pynapple addresses the concerns of eﬃciency and accuracy”

Page 9The authors wrote -This is implemented via a set of specialized object subclasses of the BaseLoader class. To avoid code redundancy, these I/O classes inherit the properties of the BaseLoader class. "From a programming perspective, the point of a base class is to avoid redundancy, so it might be better to just mention that this avoids the need to redeﬁne I/O operations in each class.

We have rephrased the sentence as follows:

“This is implemented via a set of specialized object subclasses of the BaseLoader class, avoiding the need to redeﬁne I/O operations in each subclass"

The authors wrote -"classes are unique and independent from each other, ensuring stability"How do classes being unique and independent ensure stability? Perhaps here again the misunderstanding is due to the lack of a deﬁnition of stability.

We thank the referee for the remark. We ﬁrst changed “stability” for “long-term backward compatibility”. We further added the following sentence to clarify this claim. “For instance, if the spike sorting tool Phy changes its output in the future, this would not aﬀect the “Neurosuite” IO class as they are independent of each other. This allows each tool to be updated or modiﬁed independently, without requiring changes to the other tool or the overall data format.”

The authors wrote -"Using preexisting code to load data in a speciﬁc manner instead of rewriting already existing functions avoids preprocessing errors"Here it might be helpful to use the lingo of Object-oriented programming. (e.g. inheritance and polymorphism). Deﬁning these terms for a neuroscience audience would be useful as well.

We do not think it is necessary to use too much technical term in this manuscript. However, this sentence was indeed confusing. We have now simpliﬁed it:

“[…], users can develop their own custom I/O using available template classes. Pynapple already includes several of such templates and we expect this collection to grow in the future.”

Page 10The authors wrote -"These analyses are powerful because they are able to describe the relationships between time series objects while requiring the fewest number of parameters to be set by the user."It is not clear that this makes for a powerful analysis as opposed to an easy-to-use analysis.

We have changed “powerful” with “easy to use".

Page 12"they are built-in and thus do not have any external dependencies"If the authors want to retain this, it would be helpful to explain (perhaps in the introduction) why having fewer external dependencies is useful. And is it true that these functions use only base python classes?

We have rephrased this sentence as follows:

“they are for the most part built-in and only depend on a few common external packages, ensuring that they can be used stand-alone without relying on packages that are at risk of not being maintained or of not being compatible in the near future.”

Other comments:It would be helpful, as mentioned in the public review, to frame this work in the broader context of what is needed to go from data to scientiﬁc results so that people understand what this package does and does not provide.

We have added the following sentence to the discussion to make sure readers understand:

“The path from data collection to reliable results involves a number of critical steps: exploratory data analysis, development of an analysis pipeline that can involve custom-made developed processing steps, and ideally the use of that pipeline and others to replicate the results. Pynapple provides a platform for these steps.”

It would also be helpful to describe the Pynapple so<ware ecosystem as something that readers could contribute to. Note here that GNU may not be a good license. Technically, GNU requires any changes users make to Pynapple for their internal needs to be oﬀered back to the Pynapple team. Some labs may ﬁnd that burdensome or unacceptable. A workaround would be to have GNU and MIT licenses.

The main restriction of the GPL license is that if the code is changed by others and released, a similar license should be used, so that it cannot become proprietary. We therefore stick to this choice of license.

We would be more than happy to receive contributions from the community. To note, several users outside the lab have already contributed. We have added the following sentence in the introduction:

“As all users are also invited to contribute to the Pynapple ecosystem, this framework also provides a foundation upon which novel analyses can be shared and collectively built by the neuroscience community.”

This so<ware shares some similarities with the nelpy package, and some mention of that package would be appropriate.

While we acknowledge the reviewer's observation that Nelpy is a similar package to Pynapple, there are several important diﬀerences between the two.

First, Nelpy includes predeﬁned objects such as SpikeTrain, BinnedSpikeTrain, and AnalogSignal, whereas Pynapple would use only Ts and Tsd for those. This design choice was made to provide greater ﬂexibility and allow users to deﬁne their own data structures as needed.

Second, Nelpy is primarily focused on electrophysiology data, whereas Pynapple is designed to handle a wider range of data types, including calcium imaging and behavioral data. This reﬂects our belief that the NWB format should be able to accommodate diverse experimental paradigms and modalities.

Finally, while Nelpy oﬀers visualization and high-level analysis tools tailored to electrophysiology, Pynapple takes a more general-purpose approach. We believe that users should be free to choose their own visualization and analysis tools based on their speciﬁc needs and preferences.

The package has now been cited.

**Reviewer #2 (Public Review):**
Pynapple and Pynacollada have the potential to become very valuable and foundational tools for the analysis of neurophysiological data. NWB still has a steep learning curve and Pynapple oﬀers a user- friendly toolset that can also serve as a wrapper for NWB.The scope of the manuscript is not clear to me, and the authors could help clarify if Pynacollada and other toolsets in the making become a future aspect of this paper (and Pynapple), or are the authors planning on building these as separate publications.The author writes that Pynapple can be used without the I/O layer, but the author should clarify how or if Pynapple may work outside NWB.

Absolutely. Pynapple can be used for generic data analysis, with no requirement of speciﬁc inputs nor NWB data. For example, the lab is currently using it for a computational project in which the data are loaded from simple ﬁles (and not from full I/O functions as provided in the toolbox) for further analysis and ﬁgure generation.

This was already noted in the manuscript, last paragraph of the section “Importing data from common and custom pipelines”

“Third, users can still use Pynapple without using the I/O layer of Pynapple.”.

We have added the following sentence in the discussion

“To note, Pynapple can be used without the I/O layer and independent of NWB for generic, on-the-ﬂy analysis of data.”

This brings us to an important fundamental question. What are the advantages of the current approach, where data is imported into the Ts objects, compared to doing the data import into NWB ﬁles directly, and then making Pynapple secondary objects loaded from the NWB ﬁle? Does NWB natively have the ability to store the 5 object types or are they initialized on every load call?

NWB and Pynapple are complimentary but not interdependent. NWB is meant to ensure long-term storage of data and as such contains a as much information as possible to describe the experiment. Pynapple does not use NWB to directly store the objects, however it can read from NWB to organize the data in Pynapple objects. Since the original version of this manuscript was submitted, new methods address this. Speciﬁcally, in the current beta version, each object now has a “save” method. Obviously, we are developing functions to load these objects as well. This does not depend on NWB but on npz, a Numpy speciﬁc ﬁle format. However, we believe it is a bit too premature to include these recent developments in the manuscript and prefer not to discuss this for now.

Many of these functions and objects have a long history in MATLAB - which documents their usefulness, and I believe it would be ﬁsng to put further stress on this aspect - what aspects already existed in MATLAB and what is completely novel. A widely used MATLAB toolset, the FMA toolbox (the Freely moving animal toolbox) has not been cited, which I believe is a mistake.

We agree that the FMA toolbox should have been cited. This ha now been corrected.

Pynapple was ﬁrst developed in Matlab (it was then called TSToolbox). The ﬁrst advantage is of course that Python is more accessible than Matlab. It has also been adopted by a large community of developers in data analysis and signal processing, which has become without a doubt much larger than the Matlab community, making it possible to ﬁnd solutions online for virtually any problem one can have. Furthermore, in our experience, trainees are now unwilling to get training in Matlab.

Yet, Python has drawbacks, which we are fully aware of. Matlab can be very computationally eﬃcient, and old code can usually run without any change, even many years later.

A limitation in using NWB ﬁles is its standardization with limited built-in options for derived data and additional metadata. How are derived data stored in the NWB ﬁles?

NWB has predetermined a certain number of data containers, which are most common in systems neuroscience. It is theoretically possible to store any kind of data and associated metadata in NWB but this is diﬃcult for a non-expert user. In addition, NWB does not allow data replacement, making is necessary to rewrite a whole new NWB ﬁle each time derived data are changed and stored. Therefore, we are currently addressing this issue as described above. Derived data and metadata will soon be easy to store and read.

How is Pynapple handling an existing NWB dataset, where spikes, behavioral traces, and other data types have already been imported?

This is an interesting point. In theory, Pynapple should be able to open a NWB ﬁle automatically, without providing much information. In fact, it is challenging to open a NWB ﬁle without knowing what to look for exactly and how the data were preprocessed. This would require adapting a I/O function for a speciﬁc NWB ﬁle. Unfortunately, we do not believe there is a universal solution to this problem. There are solutions being developed by others, for example NWB Widgets (NWB Widgets). We will keep an eye on this and see whether this could be adapted to create a universal NWB loader for Pynapple.

**Reviewer #2 (Recommendations For The Authors):**
Other tools and solutions are being developed by the NWB community. How will you make sure that these tools can take advantage of Pynapple and vice versa?

We recognize the importance of collaboration within the NWB community and are committed to making sure that our tools can integrate seamlessly with other tools and solutions developed by the community.

Regarding Pynapple speciﬁcally, we are designing it to be modular and ﬂexible, with clear APIs and documentation, so that other tools can easily interface with it. One important thing is that we want to make sure Pynapple is not too dependent of another package or ﬁle format such as NWB. Ideally, Pynapple should be designed so that it is independent of the underlying data storage pipeline.

Most of the tools that have been developed in the NWB community so far were designed for data visualisation and data conversion, something that Pynapple does not currently address. Multiple packages for behavioral analysis and exploration of electro/optophysiological datasets are compatible with the NWB format but do not provide additional solutions per se. They are complementary to Pynapple.